# Energy-loss return gate via liquid dielectric polarization

Taehun Kim[1], Hyungseok Yong[1], Banseok Kim[1], Dongseob Kim [2], Dukhyun Choi[3],
Yong Tae Park [4] & Sangmin Lee[1]

There has been much research on renewable energy-harvesting techniques. However, owing to increasing energy demands, significant energy-related issues remain to be solved. Efforts aimed at reducing the amount of energy loss in electric/electronic systems are essential for reducing energy consumption and protecting the environment. Here, we design an energy-loss return gate system that reduces energy loss from electric/electronic systems by utilizing the polarization of liquid dielectrics. The use of a liquid dielectric material in the energy-loss return gate generates electrostatic potential energy while reducing the dielectric loss of the electric/electronic system. Hence, an energy-loss return gate can make breakthrough impacts possible by amplifying energy-harvesting efficiency, lowering the power consumption of electronics, and storing the returned energy. Our study indicates the potential for enhancing energy-harvesting technologies for electric/electronics systems, while increasing the widespread development of these systems.

[1] School of Mechanical Engineering, Chung-Ang University, 84 Heukseuk-ro, Dongjack-gu, Seoul 156-756, Republic of Korea. [2] Aircraft System Technology Group, Korea Institute of Industrial Technology, 57 Yangho-gil, Yeongcheon-si, Gyeongbuk-do 38822, Republic of Korea. [3] Department of Mechanical Engineering, Kyung Hee University, 1732 Deogyeong-daero, Giheung, Yongin, Gyeonggi 446-701, Republic of Korea. [4] Department of Mechanical Engineering, Myongji University, 116 Myongji-ro, Cheoin-gu, Yongnin, Gyeonggi 17058, Republic of Korea. Equal contribution: Taehun Kim, Hyungseok Yong. Correspondence and requests for materials should be addressed to Y.T.P. (email: ytpark@mju.ac.kr) or to S.L. (email: slee98@cau.ac.kr)

In recent decades, there has been a significant growth in interest for preventing and responding to issues associated with the increasing demands of electricity production, including environmental, economic, resource, and safety issues[1–10]. Accordingly, the search for sustainable energy-harvesting methods, by which electrical energy is generated from alternate sources, is an emerging field of research. To date, many renewable energy-harvesting techniques utilizing photoelectric[11–13], piezoelectric[14–16], thermoelectric[17–19], pyroelectric[20–22], and triboelectric effects[23–25] have been proposed to facilitate energy-related technology and systems. The principles considered have become well established in recent years. In addition to the traditional electricity generation methods, the recent eco-friendly energy-harvesting methods generate significant electricity for society, widen cutting-edge technology applications, and improve the quality of life. Although the amount of electricity generated has increased, there is a continuous growth in demand for electric energy owing to widespread development and deployment of electronic devices. Hence, innovative science and engineering methods must be considered to address future electric power shortages. In general, all electric/electronics systems and energy-generating methods have inevitable energy loss when operated. It is thus necessary to consider the causes of energy loss and methods to recapture energy loss as available energy.

There are several types of energy loss, including loss caused by thermal[26,27], vibrational[28,29], acoustic[30,31], and electromagnetic[32,33] effects. Alternating electric field, which are induced by all electric systems, affect the adjacent dielectric material, changing its polarization direction. Thus, some of the input energy is used in heating the dielectric material, causing energy loss in the form of heat dissipation. In other words, materials unrelated to the electric system are polarized by electrostatic induction, thereby decreasing the electric efficiency due to dielectric loss[34,35]. Many methods have been studied to reduce dielectric loss;[36–40] however, no fundamental approach has yet been proposed for converting dielectric loss into available electric energy for the operation of electronic systems. This suggests an opportunity to design a highly efficient mechanism for dielectric loss reduction by introducing an energy-loss return process. To develop this process, it is necessary to select a stable material that can transfer substantial amounts of energy without additional energy loss. Among the possible materials, liquid dielectrics provide significant benefits in that they offer high dielectric strength, can serve as a refrigerant, prevent corona discharge, and suppress electric arcing. These characteristics are well suited for energy transfer, while maintaining electric stability, and provide the basis for an efficient energy return system. To minimize the dielectric loss, an investigation of the relationship between liquid dielectric properties and energy transfer is performed. The result is an increase in both the available electric power and the efficiency of the overall system.

Through studies of electric field effects on liquid dielectric polarization, we investigate the energy-loss return gate (ELRG) as a novel method for converting dielectric loss into available energy. The primary mechanism is demonstrated using a triboelectric generator (TEG), which generates electrical potential in the form of a pulsed AC signal. We have successfully designed an amplified TEG with an energy-loss return system that can amplify electrical peak power by 350%, and charging performance by 240%, with equivalent input energy. Using electrostatic simulation, we show how the charge and electric potential of the system are transferred from one medium to another. Furthermore, we show how the liquid dielectric polarization effect, due to the molecular polarization characteristics, can be used in the ELRG to harvest the energy loss from a wide array of electronic devices used in daily life. These electric/electronic systems can be operated more efficiently through the cooling effect arising from decreased power consumption. Most importantly, this approach of using returned energy loss as an available electric energy source can be implemented without a complex fabrication process. The method is simple and intuitive and provides a means for addressing the problem of energy loss and electric/electronic efficiency in a given system, while amplifying energy-harvesting efficiency, reducing the power consumption of electronics, and storing the retrieved energy for use in another system. The ELRG is an innovative approach in the sense that it can enhance many other superior energy-harvesting technologies, in addition to improving electronics efficiency. Thus, adapting this methodology will aid in the development of energy-harvesting research as well as other energy-related engineering disciplines.

## Results

**Design of energy-loss return framework.** Figure 1 shows a schematic of the energy transfer process, illustrating the electricity generation, the loss mechanisms, and the ELRG. Compared with conventional energy-harvesting methods, which operate under specific conditions, the ELRG is a fundamental technology and novel energy-harvesting method for transferring energy loss to available energy. The ELRG configuration and step-by-step mechanism are illustrated schematically in Fig. 2. The configuration of the ELRG can be classified into four electric regimes: the AC electric system, the dielectric loss layer, the liquid dielectric layer, and the electrode. The AC electric system can represent all commonly used electricity-related devices. The electric fields generated by these devices affect adjacent materials

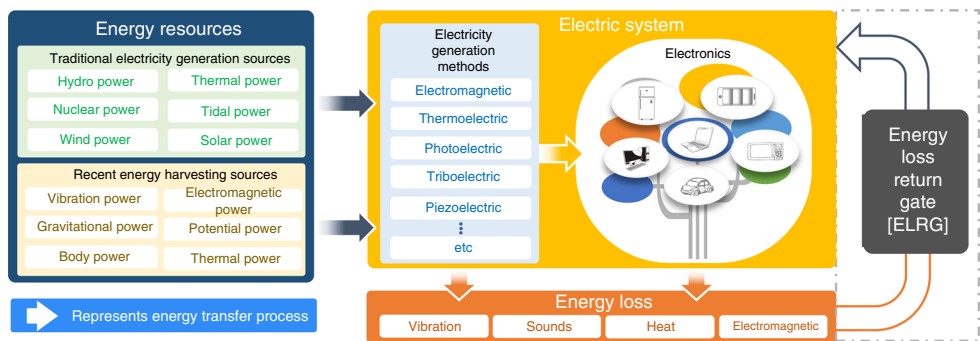

**Fig. 1** ELRG is conceptualized to convert energy loss to available energy. Lines are added to show the energy transfer process in electric/electronics systems

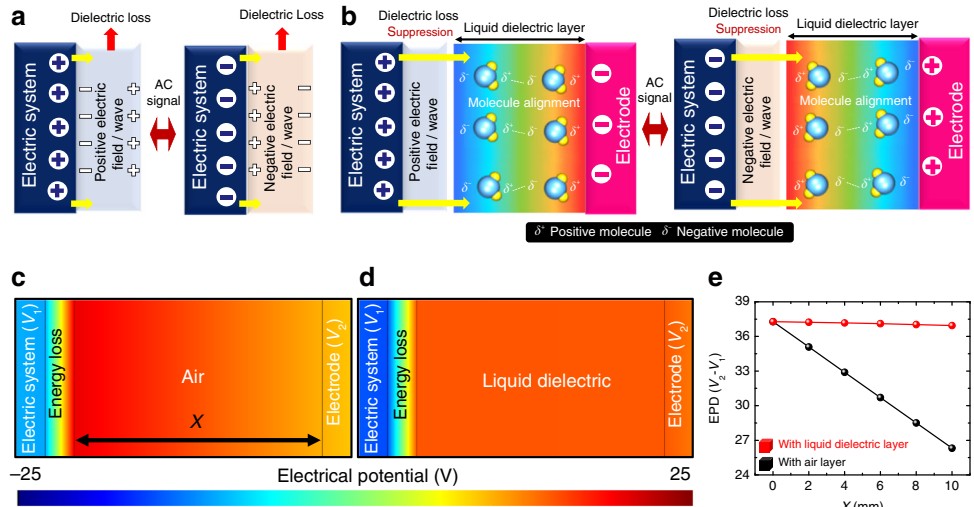

**Fig. 2** ELRG is available with a liquid dielectric polarization. **a** The existence of a liquid dielectric layer determines the energy transfer process. Electric/electronics systems dissipate electromagnetic energy without a liquid dielectric layer. **b** Liquid dielectric layer scavenges the dissipated electromagnetic energy. **c** Electrostatic simulation resultants. Insets show the degree of generated electric potential with air layer. **d** The degree of generated electric potential with liquid dielectric layer. **e** Electrostatic potential differences between electric system and electrode. EPD is more stable when liquid dielectric layer is integrated

that are not related with device operation, for example, a device case, support, and coatings. Some of these dielectric materials are easily polarized, with the direction of polarization continuously changing in accordance with the input AC frequency of the electric system. As a result, some energy is dissipated as heat, resulting in dielectric loss. In other words, if there is an AC power-based electric system, the electric field inherently induces a dielectric loss that dissipates energy as heat (Fig. 2a). To prevent dielectric loss, we introduce a liquid dielectric layer to facilitate the ELRG process. When there is a liquid dielectric layer beside the dielectric loss layer, the electric field from the AC electric system is directly transferred to the liquid dielectric layer owing to its high relative permittivity, so that there is no significant energy dissipation in the dielectric loss layer. This is because liquid dielectric materials are more easily polarized than the dielectric loss layer materials (Supplementary Fig. 1). Consequently, the dipole moment in the liquid dielectric is aligned, storing polarization energy. The liquid dielectric dipole alignment enables transfer of polarization energy to the electrode, adjusting the molecular alignment direction. Finally, the electrode inside the liquid dielectric is defined as an electric potential inducing medium, because the electrode could generate charge transfer via electrostatic induction (Fig. 2b). In summary, this illustrates the ELRG process by which energy loss is returned as available electric energy. Details of the mechanism, process, and effects are discussed in the following sections.

**Electrostatic simulation for the ELRG fabrication**. We performed electrostatic 2D simulations using COMSOL FEM simulation software to calculate the electric potential for material relative permittivity in an ideal environment. Detailed assumptions and the simulation process are described in Supplementary Fig. 2. We define four layers comprising the electric/electronic system and the ELRG: the electric system ($V_1$), the first dielectric layer (dielectric loss layer), the second dielectric layer (liquid dielectric layer), and the ELRG electrode ($V_2$). We assume a hypothetical layer called an "energy loss layer," which has an identical amount of surface charge density with

different signs ($|\sigma_h^+| = |\sigma_h^-|$), to focus on the effect of the energy loss transfer process. Through these logical assumptions, we can compare the effects of the ELRG system with and without the liquid dielectric layer. The fundamental mechanism of the ELRG system is to generate electrostatic potential in the ELRG electrode via the liquid dielectric layer. Thus, by calculating the distribution of electrostatic potential inside the ELRG system, it is possible to demonstrate the principles of the ELRG system by varying the material properties in the second dielectric layer.

Figure 2c, d shows the distribution of electric potential, depending on the relative permittivity of second dielectric layer; all parameters used in this simulation are listed in Table 1 and Supplementary Fig. 2. In the general case, the electric/electronic system operates in the air ($\varepsilon_r = 1$). Since the air is not easily polarized by the electric field, owing to its low relative permittivity, the electric potential in the air decreases rapidly as a function of distance away from the energy loss layer. In this case, the polarization energy stays in the energy loss layer, dissipated as heat, and transfer of the energy loss to the electrode is minimal (Fig. 2c). However, when a liquid dielectric layer of high relative permittivity ($\varepsilon_r = 80$) is located next to the dielectric loss layer, polarization energy is transferred to the liquid dielectric layer by the dipole alignment. The distribution of electric potential is thus more uniform in the liquid dielectric layer in comparison to the air layer; therefore, the liquid dielectric layer can transfer electric potential to the ELRG electrode (Fig. 2d). Figure 2e shows a plot of the electric potential difference (EPD) between the electric system ($V_1$) and the ELRG electrode ($V_2$) as a function of distance ($X$), which represents the distance between the first dielectric layer and the ELRG electrode. When there is an air layer between the electric system ($V_1$) and the electrode ($V_2$), the EPD decreases rapidly with distance ($X$). However, with a liquid dielectric layer between the electric system ($V_1$) and the electrode ($V_2$), the EPD is maintained as a function of distance ($X$). The effect of a liquid dielectric in the ELRG system can thus be successfully analyzed by electrostatic simulation under ideal conditions, and can help clarify the theoretical mechanism of the ELRG system.

**Table 1 Parameters for the electrostatic simulation**

| Parameter | Value |
|---|---|
| Surface charge density of energy loss layer, $\sigma_h$ | $1\ \mu C \bullet m^{-2}$ |
| Relative permittivity of energy loss layer, $\varepsilon_{loss}$ | 3 |
| Relative permittivity of air, $\varepsilon_0$ | 1 |
| Relative permittivity of de-ionized water, $\varepsilon_{liquid}$ | 80 |
| Ambient temperature, $T$ | 20 °C |

**Energy-loss return process and effect**. To analyze the ELRG system, we introduce a simple TEG to study the AC electric performance. Invented by the group of Professor Wang in 2012, both TEG and triboelectric nanogenerator (TENG) are energy-harvesting methods that convert mechanical energy into electric energy[23–25]. Triboelectrification is a surface-charging process in which electrostatic potential is generated by a continuous process of contact and separation between materials. Having neither high cost nor complex fabrication, it can be integrated to harvest waste mechanical energy. Moreover, the TEG mechanism is also a reliable input energy source, because electric output frequency and amplitude can be easily adjusted. For this reason, TEGs are employed as controllable electric input sources to develop strategies for investigating ELRG mechanisms.

As shown in Fig. 3, the ELRG with a TEG is set up in five parts: the triboelectrification layer, the first electrode, the dielectric loss layer, the liquid dielectric layer (water layer), and the second electrode. Details of the experimental method are illustrated in Supplementary Fig. 3. The triboelectrification layer and the first electrode are considered as a general AC electric system, which generates dielectric loss from the electric field. At first, the negatively charged triboelectrification material and the electrode are in an equilibrium state without generating electric charge in the first electrode (Fig. 3a). Because of contact electrification between the triboelectrification material and the electrode, the first electrode becomes positively charged, while the EPD between the first and second electrodes drives the stream of free electrons. As a result, electric current flows between the two electrodes in the direction illustrated in Fig. 3b. After the triboelectrification material separates from the first electrode, current flows in the opposite direction between the two electrodes. It clearly comes as no surprise that both the triboelectrification layer and the first electrode return to an equilibrium state (Fig. 3c). Because the TEG system is an AC power-based system, as discussed above, there is a continuously changing direction of polarization in the dielectric loss layer, inducing dielectric loss. Some parts of the material unrelated to the triboelectric processes are polarized by electrostatic induction from the electric field; hence, the TEG is an appropriate mechanism to demonstrate and analyze the ELRG system.

To demonstrate the ELRG mechanism with the TEG, we fixed all experimental conditions except for the presence of the liquid dielectric layer (Fig. 3d-f). If there is no liquid dielectric layer between the dielectric loss layer and the second electrode, the dielectric loss layer dissipates some energy as a heat (Fig. 3a -c). If there is a liquid dielectric layer, the polarization energy from the AC electric system is transferred to polarization energy of the liquid dielectric (Fig. 3d-f). This polarization energy is stored in the liquid dielectric layer and affects the second electrode electrostatically. As a result, an additional electric potential is generated in the second electrode. The sign of the charge on the second electrode is different than that on the first electrode, in accordance with charge alignment, as illustrated in Fig. 3d-f.

Consequently, compared to the TEG mechanism without the liquid dielectric layer, the ELRG system enhances the EPD between the first and second electrodes, for equivalent input conditions.

To verify the electrical performance of the ELRG, we selected the first electrode as a probe and the second electrode as a ground. As plotted in Fig. 4a, b, with the aid of the ELRG system, the open-circuit voltage ($V_{oc}$) is increased by a factor of 1.58 (measured to be −24.4 V and −38.6 V for the liquid dielectric layer absent and present, respectively). The short-circuit current ($I_{sc}$) is increased by a factor of 1.69 (measured to be −2.4 and -4.06 μA, respectively). To confirm this enhanced performance, we interchanged the probe and ground for the liquid dielectric layer, and found no significant difference with the original measurements, except for the change in electric sign. This result confirms that the electric potential between the first and second electrodes is amplified with the aid of the ELRG system by the additional electric potential in the second electrode. Moreover, we measured the electrical power output under various external load resistances ranging from 10 kΩ to 1 GΩ. Under these load resistances, the average peak voltage and current measurements are shown in Supplementary Fig. 4. The maximum peak power of the TEG with the liquid dielectric layer is increased by a factor of 3.5 (measured to be 8.3 μW and 28.9 μW, respectively, under a load resistance of 20 MΩ, compared with the TEG without the liquid dielectric layer (Fig. 4c). To further indicate the effectiveness of the ELRG, a 1-μF capacitor is connected to the TEG in both the presence and absence of the liquid dielectric layer. The charging rate of the capacitor with the liquid dielectric layer is faster (by a factor of 2.4) than the original TEG without the liquid dielectric layer (Fig. 4d). This demonstrates that when the ELRG is integrated into the TEG, it is possible to generate greater electric power and efficiency than without the ELRG by scavenging the energy loss. Furthermore, this ELRG system provides an easily applicable mechanism and a low fabrication cost, in addition to high efficiency. Hence, this novel method has the potential for providing effective solutions in many energy-related fields.

**Marriage of ELRG system and electronics**. We propose a practical energy-harvesting method, in which energy loss is converted to useful electric energy, while reducing power consumption, for a wide range of electric/electronic devices. Fig. 5a, b illustrates the step-by-step mechanism of the energy-harvesting process utilizing the ELRG system. When an electronic device is turned on, the electric field causes energy loss and heat dissipation due to the continuously changing dipole alignment, as discussed in the above sections. For electronic devices, this has an adverse impact that can include reduction in efficiency, shortened lifetime, and other problems. By integrating ELRG with common electronic devices, two potential benefits can be achieved—harvesting of energy loss, and reduced power consumption. For this purpose, we introduce tap water as the liquid dielectric material, to enable a sustainable ELRG process. Tap water is used owing to its abundance as a resource, non-toxicity, high relative permittivity, high specific heat, high thermal conductivity, and low viscosity. Water absorbs the electric fields generated by electronics owing to its high relative permittivity, and as a result it is possible to convert the energy loss into polarization energy inside the water. Moreover, water absorbs heat from the electronics, maintaining a lower device temperature than under normal operating conditions. Therefore, it is also possible to decrease the power consumption of electronics by using the ELRG. Water molecules aligned with the AC electric field induce alternating charge transfers on the connected electrodes by electrostatic

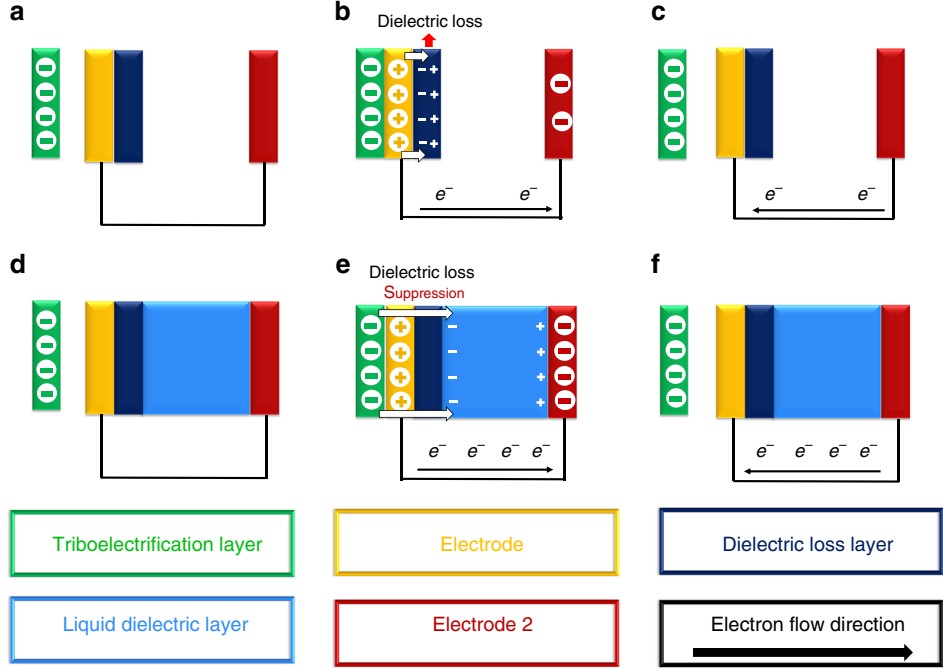

**Fig. 3** Conformation and process of triboelectric generator. **a** Equilibrium state of a TEG without liquid dielectric layer. **b** Contact state without liquid dielectric layer. **c** separation state without liquid dielectric layer. **d** Equilibrium state of a TEG with liquid dielectric layer. **e** Contact state with liquid dielectric layer; and **f** separation state with liquid dielectric layer

induction. Hence, by combining the ELRG with electric components such as rectifiers, capacitors, and transformers, the electric energy recovered from the energy loss can be stored in a battery to operate other electronic devices (Fig. 5b).

The performance of the ELRG applied to electronic systems is demonstrated in Fig. 5c-e, with experimental geometric parameters (Supplementary Fig. 5), to clarify the effectiveness of the ELRG when used with electronic devices. As shown in Supplementary Fig. 6, the experimental components consist of four parts: an electric motor covered with urethane film as an electric device, urethane film with a thickness of 0.3 mm acting as a dielectric loss layer, water in a PET case as a liquid dielectric layer, and a wattmeter with 9 mW resolution to measure the electric power consumption. Without water in the PET case, the electric device consumes 5.17 watts for an hour (W·h) while the device is operating. As the PET case is filled with water, the power consumption of the device decreases to 5.11 W·h under equivalent experimental conditions (Fig. 5c, Supplementary Fig. 6a-c). Because some electric components/materials perform more efficiently under lower ambient temperatures, these results suggest that the water acts as a liquid dielectric material that functions as a coolant, and lowers the power consumption[41,42]. The reduced power consumption rates regarding various thermal conductivities are shown in more detail in Supplementary Fig. 6e, f. When the ELRG energy-harvesting circuit is installed for electric energy storage, the capacitor (1000 µF) is charged, generating considerable electric performance and exceeding the wattmeter resolution (Fig. 5d, Supplementary Fig. 6c, d). Moreover, the recovered power level can be estimated as: Recovered power from ELRG per total power consumption = 148 V × 125 µA per 5110 mW = 0.00359 = 0.36%, and the power density can be calculated as: Recovered power from ELRG per volume of liquid dielectric = 148 V × 125 µA per 0.001 m$^3$ = 18.583 W·m$^{-3}$. However, there is no increase of power consumption, which is measured to be 5.11 W·h in both cases. This

experimental result agrees with the theoretical analysis of ELRG systems, for which energy loss is converted to available electric energy while leaving the operation of the electric device undisturbed. To further verify the principle of the ELRG, we measured the electric power under various external load resistances, taking the inherent ELRG impedance into account. As shown in Fig. 5d, the ELRG generates high output voltage on a scale of hundreds of volts while generating very low current on a scale of microamperes. This is the result of electrostatic induction, such as that exhibited by triboelectric and piezoelectric generators. Owing to this reason, the maximum peak power from both the ELRG process and a TEG is measured under a load resistance of 20–30 MΩ (Fig. 5e). These results demonstrate that the impedance of the ELRG mechanism is driven by electrostatic induction, which is similar to that of TEG processes. The effectiveness of the ELRG has thus been successfully demonstrated by these experiments, opening up the potential of ELRG for future practical consideration.

**Application fields with ELRG**. Figure 6a illustrates some applications of the ELRG energy-harvesting process in our daily lives. As discussed in the above sections, there is unavoidable energy loss from electronic devices such as laptops, cell phones, and computers. To scavenge energy loss from the various electronic devices, we introduce and implement an energy-harvesting model, utilizing the ELRG system. Electrical components such as rectifiers, capacitors, and regulators are applied to each device for energy-harvesting (Supplementary Fig. 7). As a result, with the aid of the ELRG process, it is possible to convert the energy loss to available electricity, which can then be applied to other electric systems. Figure 6b shows a demonstration of this application in an actual environment with the following components: ELRG 1 connected to a laptop, ELRG 2 connected to a cell phone, and an energy-harvesting circuit. The detailed electric outputs are shown

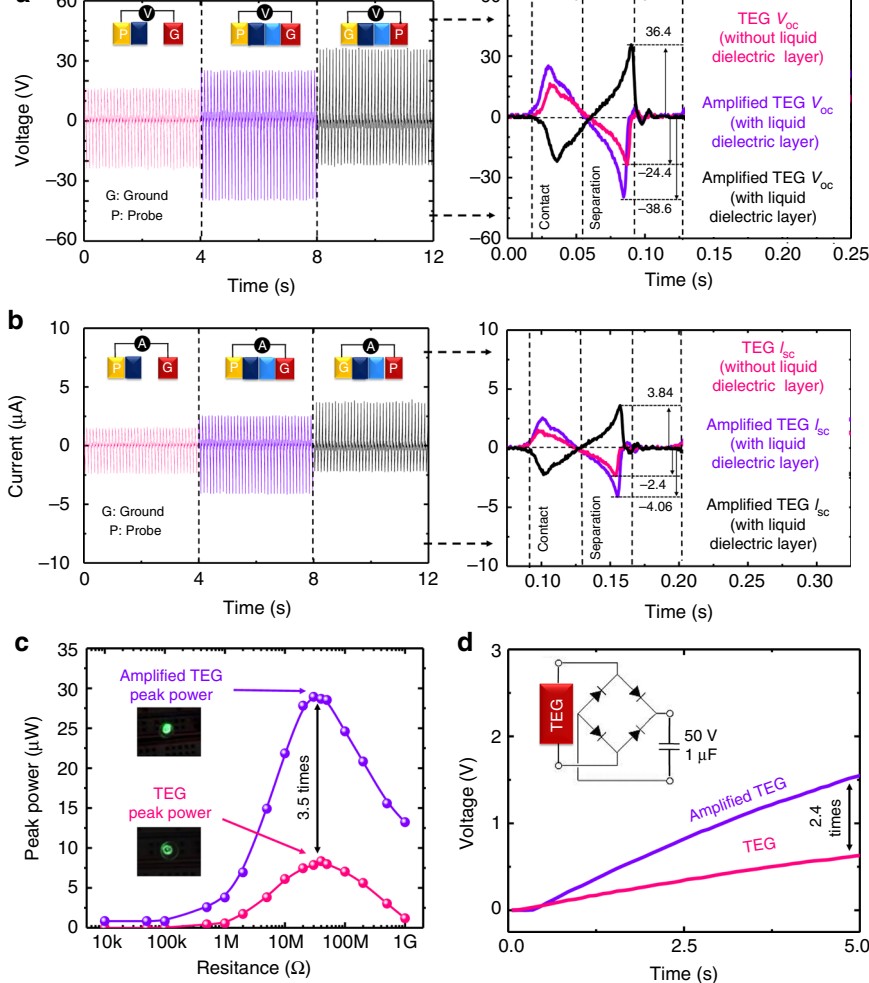

**Fig. 4** Effects of ELRG are demonstrated through a triboelectric generator for experimental validation. All electric performances are measured under a 9 Hz TEG. **a** Comparative analysis of open-circuit voltage ($V_{oc}$). Open-circuit voltage without ELRG (pink lines); with ELRG (both purple and black lines). **b** Comparative analysis of short-circuit current ($I_{sc}$). Short-circuit current without ELRG (pink lines); with ELRG (both purple and black lines). **c** Peak power performance of TEG under load resistances. Inset pictures show that TEG power with ELRG lighten an LED more brightly. **d** Capacitor charging performances of TEG

in Supplementary Fig. 8. When an ELRG energy-harvesting unit is connected to the energy-harvesting circuit, the charge storage capacitor (1000 μF) in the circuit is successfully charged by the ELRG power units (Supplementary Fig. 9). As demonstrated in Fig. 6c, a discharged Bluetooth mouse starts working after connecting it to the charged energy-harvesting circuit, operating for a few seconds. Furthermore, as illustrated in Fig. 6d, a widely used Bluetooth keyboard without a battery can be operated for tens of seconds by the ELRG system, where a 6000 μF capacitor is used in this experiment (Supplementary Movie 1).

In addition, we propose a self-powered wire-sensing system with an ELRG mechanism. As illustrated in Fig. 6e, the basic motive for this mechanism is to detect electric signals utilizing electric energy generated by an ELRG. To demonstrate the self-powered mechanism, we scavenge energy loss from electronic devices that are integrated with ELRG. When the electric energy from ELRG is supplied to a sensing system, we can detect electric signals in electrical components. Figure 6f shows a demonstration of the sensing system with three power lines—ELRG 1 connected to device 1, ELRG 2 connected to device 2, and ELRG 3 connected to device 3. To visualize the sensing system, we integrate light-emitting diodes (LEDs) in the sensing system, which is composed

of rectifiers and capacitors (10 μF; Supplementary Fig. 10). The respective capacitors are instantaneously charged because of electric energy supplied from the ELRG system, lighting up LEDs connected to the capacitors. When device 1 is turned off, the LED connected to ELRG 1 is turned off while the connected capacitor is discharged. However, when device 1 is turned on, the capacitor is charged, lighting up the LED again (Fig. 6g and Supplementary Movie 2).

## Discussion

Previous research in energy-related fields has led to the objective of lowering the energy loss of electric systems, and to operate these systems at higher efficiency. However, there has been little study of energy-generating methods with consideration of dielectric loss. In this paper, we successfully develop a new technology called the ELRG, which can convert energy loss to available electric energy, opening up new opportunities in energy-related engineering fields. This research enables a reduction of energy loss in electric systems caused by dielectric loss, via a dielectric loss reduction method. To pursue the fundamental development and investigation of the ELRG, we introduced a

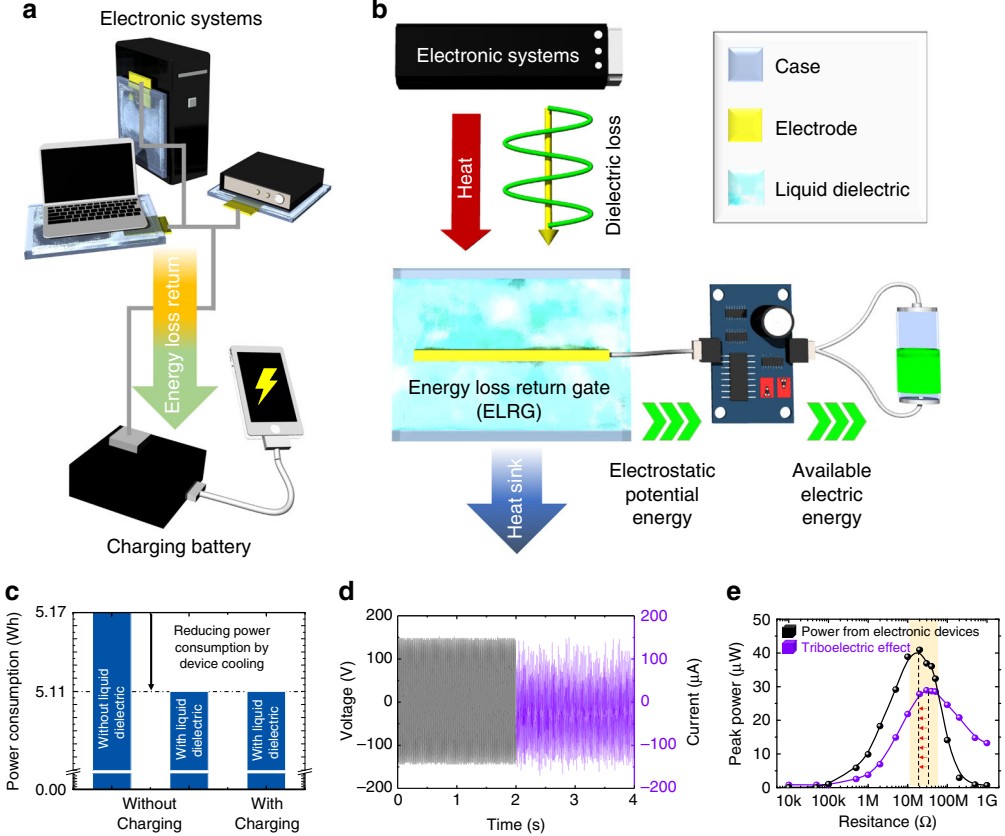

**Fig. 5** Operation schemes and electric performances from ELRG process. Electric device is plugged into a 60-Hz AC input electrical source. **a** Configuration of ELRG for practical application. **b** Operation scheme of ELRG with electric components. **c** Power consumption measurements from wattmeter. Without liquid dielectric/without charging (left); with liquid dielectric/without charging (middle); with liquid dielectric/with charging (right). Dotted black line represents no difference. **d** ELRG outputs from the electric device. Open-circuit voltage (left plot); short-circuit current (right plot). **e** Peak power plots under load resistances. Measurements show the inherent impedance of ELRG and TEG. The difference is highlighted by dashed black lines and red arrows

liquid dielectric material, which has high relative permittivity and other significant advantages, to transfer the electrostatic potential energy. The mechanism was demonstrated using a TEG, and electrostatic simulations were implemented using COMSOL software. We further demonstrated the use of the ELRG system in electronics, with everyday electronic devices, to confirm the practical feasibility of the approach. The results show the innovation of the method, and indicate the potential for improving overall energy systems by means of the following three salient features. First, if the ELRG is integrated with energy-generating methods, the returned energy loss can amplify the energy-harvesting efficiency under the same input conditions. Second, the ELRG process decreases the temperature of electric systems because of the high thermal conductivity of liquid dielectric materials, which lowers power consumption of the electronics without adverse effects. Third, the returned energy from dielectric loss can be stored in a battery, which can then function as an energy source for another electronic device. Our results highlight the energy transfer processes that occur within every electric system and approach the energy-related issues from a scientific and engineering perspective. The advantages and benefits of the ELRG process are not limited to a specific application field, but hold promise for application to a wide range of energy-related fields.

## Methods

**Impact of liquid dielectric material**. To verify the effects of liquid dielectric materials, in consideration of their relative permittivity, we conduct two

experiments to measure electric potential generated by the ELRG process, with regard to relative permittivity and ionic concentration, respectively.

In general cases, the dielectric constant is decreased as the input frequency is increased. In this study, the ELRG is investigated in the very low frequency range, up to 60 Hz. Thus, we assume that the liquid dielectric constant is not reduced by input frequency for the ELRG process. First, Supplementary Figure 1a shows the results of induced electric potential in the electrode with four different liquid dielectric materials: under room temperature (20 °C), water ($\varepsilon_r$ = 80.1), n-Hexane ($\varepsilon_r$ = 1.89), Isopropanol ($\varepsilon_r$ = 20.33), and Methanol ($\varepsilon_r$ = 32.63). As the relative permittivity of the liquid dielectric material increases, the electric performance of the ELRG also increases inducing a higher potential in the electrode until the electric performance gradually reaches saturation. Second, we investigate the ionic effect, dissolving NaCl ions into DI-water solvent, to confirm the operating principle of the ELRG process under identical experimental conditions (Supplementary Figure 1b). Unlike with relative permittivity, there is no difference in the electric performance of the ELRG, maintaining the electric potential in the electrode until the concentration of NaCl ions reaches 5 M. From this analysis of liquid dielectric materials, we have demonstrated the theory of the ELRG mechanism, showing the effect of the liquid dielectric layer (Supplementary Fig. 1c and d).

**Simulation conformation for COMSOL**. As discussed in the manuscript, when polarization occurs in the first dielectric layer (dielectric loss layer), the energy from the electric system is dissipated as dielectric loss. However, if the polarization occurs in the second dielectric layer (liquid dielectric layer), some energy can be transferred as electrostatic potential to the electrode of the ELRG through molecular alignment. We conducted an electrostatic simulation to investigate the electric properties, which are particularly critical in integrating the ELRG with electric systems. Supplementary Fig. 2 shows the schematic diagram and results of the simulation. The assumptions are as follows. First, we define four layers comprising the electric system and the ELRG, namely: the electric system ($V_1$), the first dielectric layer (dielectric loss layer), the second dielectric layer (liquid dielectric layer), and the ELRG electrode ($V_2$). Second, there is a

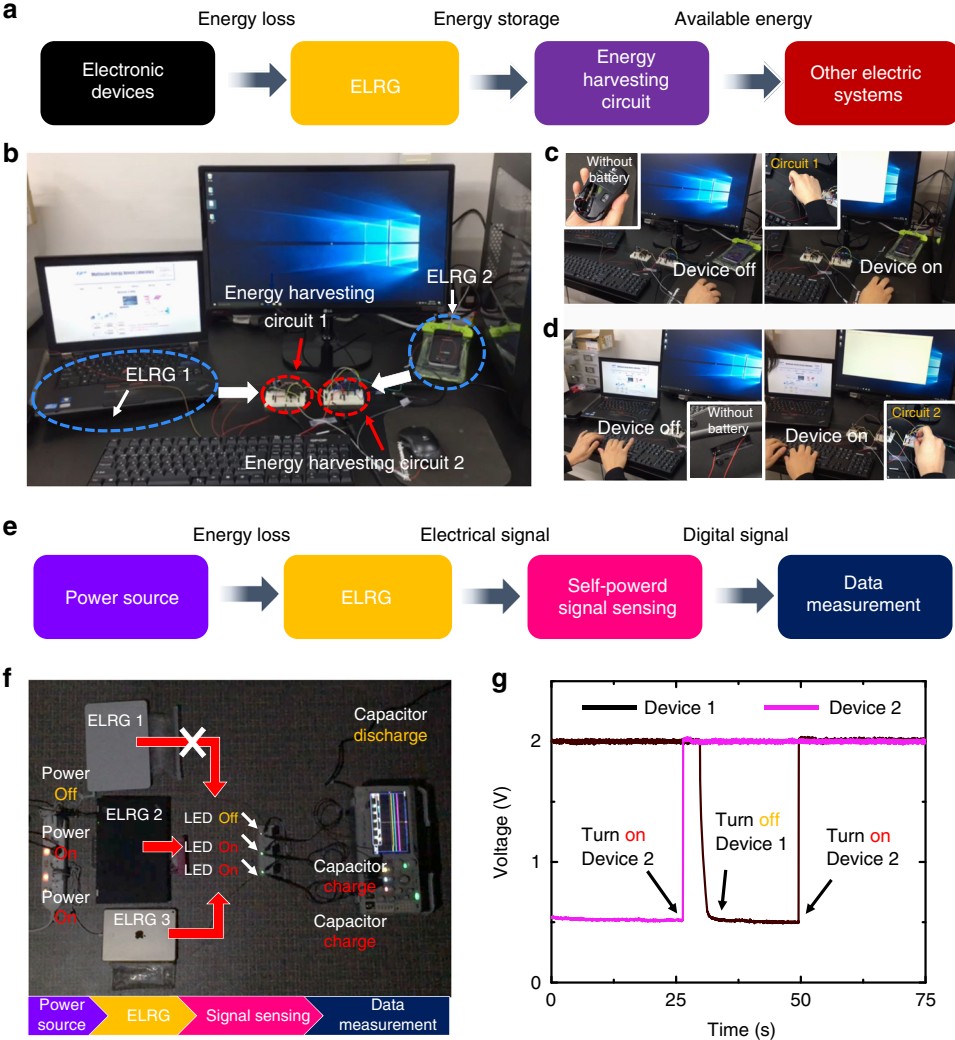

**Fig. 6** ELRG is applicable in our everyday environment. All electric devices are plugged into a 60-Hz AC input electrical source. Detailed electric outputs are given in Supplementary Figs. 8–9. **a** Integration of ELRG in individual connection for constructive energy-harvesting. **b** Optical picture of the integrated ELRG system. Two ELRG outputs are connected to an energy-harvesting circuit (from cell phone and laptop). **c** Discharged Bluetooth mouse is operated by a connected energy-harvesting circuit. **d** Discharged Bluetooth keyboard is operated by the connected energy-harvesting circuit. **e** Schematics illustrating self-powered wire-sensing conformation. **f** Optical picture of the self-powered wire-sensing ELRG system. Three ELRG outputs are detected by capacitor circuits and LEDs. **g** Plots showing electric potential variation in the connected capacitors

certain amount of surface charge density ($\rho$) in the electric system (first electrode) while the electric system operates. Third, the medium is an infinite plane, hence the volume charge density of the system ($\rho$) can be replaced by a surface charge density ($\sigma$). Finally, the charge density is continuously generated, while resulting in the generation of an electric field ($E$) and causing the materials to be polarized ($P$). By combining the above assumptions, we compared three primary values, electric field variation, polarization quantity, and electrostatic potential, to investigate the effect of the liquid dielectric layer within the ELRG process.

Supplementary Fig. 2a illustrates the simulation results of the ELRG process, where the electric system discharges some electric field from the electric loss layer. In this simulation, because the dielectric loss is generated by polarization in the first dielectric layer, we introduce a hypothetical layer called the "energy loss layer" with a surface charge density ($\sigma_h$). In other words, we assume that the energy loss layer is hypothetically polarized as two identical amounts of charge with different signs ($|\sigma_h^+| = |\sigma_h^-|$), distributed across the dielectric loss layer. When air is selected as the second dielectric layer, only the first dielectric layer is polarized while generating dielectric loss. However, when the liquid dielectric is introduced in the second dielectric layer, the second dielectric layer is more polarized than the first dielectric layer due to its high permittivity (Supplementary Fig. 2b, c). In addition, we can observe that the introduction of the liquid dielectric layer decreases the reduction in electric potential (Supplementary Fig. 2d).

Together, these results show that energy loss can be stored as a molecular alignment of the liquid dielectric in the second dielectric layer, resulting in an electrostatic potential. To investigate the electrostatic potential in the electrode of the ELRG, we calculate the EPD between the electric system ($V_1$) and the ELRG electrode ($V_2$) to investigate the charge transfer process in the ELRG (Supplementary Fig. 2d).

**Fabrication of the TEG.** The TEG is implemented by a surface-charging phenomenon, in which the continuous contact and separation between two materials generates a flow of free electrons in the electrodes. As illustrated in Supplementary Fig. 3, our TEG consists of five components: a triboelectrification layer, a first electrode, a dielectric loss layer, a liquid dielectric layer (water layer), and a second electrode. The triboelectrification layer is composed of a 5 cm-diameter and 3 mm-thick PMMA plate, covered with a 0.08 mm-thick PTFE film. Al-tape, of 5 cm-diameter with 100 μm-thickness, is employed as the first electrode material. To stabilize the status of first electrode during the TENG process, as well as to enable the ELRG mechanism, the first electrode is taped on a 10 cm-diameter and 3 mm-thick PMMA cylinder. It is worth nothing that, due to the triboelectric regime, the PMMA cylinder is not directly related to the triboelectric generation process. Rather it behaves as a dielectric loss layer, decreasing the energy-generation efficiency. In addition, we use tap water (20 °C) for the liquid dielectric layer, which generates electrostatic potential reducing dielectric loss, maintaining consistency

and practicability in the TEG experiments. On the basis of the electrostatic potential energy generated in the liquid dielectric layer from the energy loss of electric system, additional energy can be harvested at the second electrode, which is Al-film with an area of 1 cm$^2$ and thickness of 200 μm.

**Parameter study for the ELRG design**. Because the liquid dielectric layer plays a significant role in the ELRG system, there are important factors such as size, volume, and location in developing the ELRG design. The realistic electric field ($E^*$) and realistic polarization ($P^*$) are mainly related to the geometry of the liquid dielectric. Therefore, experiments with main parameters are performed under various conditions. Supplementary Fig. 5a shows a schematic, with the primary variables to be considered: diameter ($d$), height of layer ($h$), and distance ($L$) between the layer and the electronic device. In these experiments, PMMA cylinders, which have different diameters ranging from 5 to 15 cm, and heights ranging from 0 to 20 cm, were used to control the electrically affected area and volume of the liquid dielectric. As the area of the liquid dielectric increases with controlling the diameter of the cylinder, the electric potential of the ELRG also increases, generating the higher electric output. In addition, as the volume of liquid dielectric expands under equivalent sectional area, we can observe the increase in electrical output and gradual saturation, by increasing the height of the liquid dielectric (Supplementary Fig. 5b). When the distance between the liquid dielectric and the electric device increases, owing to low permittivity of air, it is hard to transfer the polarization energy to the liquid dielectric material (Supplementary Fig. 5c). As a result, it is necessary to maximize the liquid dielectric area with the proper volume, and to minimize the distance between the liquid dielectric and the electric device for optimized ELRG performance.

**Impact of thermal conductivity**. To verify the effects of cooling effect of ELRG, in consideration of various thermal conductivity of liquid dielectrics, we conduct experiments to measure the power consumption while operating the ELRG process. First, Supplementary Fig. 6e and f show the results of reduced power consumption in the electric device with four different liquid dielectric materials: under room temperature (20 °C), water (0.591 W·m$^{-1}$·K$^{-1}$), n-Hexane (0.1203 W·m$^{-1}$·K$^{-1}$), Isopropanol (0.154 W·m$^{-1}$·K$^{-1}$), and Methanol (0.203 W·m$^{-1}$·K$^{-1}$). As the thermal conductivity of the liquid dielectric material increases, the reduced power rate of the ELRG also increases.

**Data availability**. The data that support the findings of this study are available from the corresponding author upon reasonable request.

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

## Acknowledgements

This work was supported by the National Research Foundation of Korea (NRF) grant funded by the Korean government (MSIP; No. 2016R1A4A1012950) and the Nano-Material Technology Development Program through the NRF funded by the Ministry of Science, ICT and Future Planning (NRF-2016M3A7B4910532) and the Korea Institute of Industrial Technology (KITECH) as 432 "Research source technique project (EO-17-0020 and EO-17-0027)".

## Author Contributions

T.K. and S.L. discovered and conceptualized the phenomenon. T.K., H.Y., Y.T.P., and S.L. designed the project. T.K., H.Y., B.K., and D.C. conducted the experiments. H.Y. and

D.K. performed the COMSOL simulation. D.K. and D.C. commented on the manuscript. All authors wrote the manuscript and analyzed the data. Y.T.P. and S.L. supervised the whole project.

## Additional information

**Competing Interests:** The authors declare no competing interests.

