## [Peer Review File · Nature Communications]

Reviewer #1 (Remarks to the Author):

I think the main contribution of this paper is a way to concentrate the polarization effect of an alternating electric field by using a liquid dielectric layer. This is then combined with a triboelectric generator (TEG) for energy scavenge. The combination is clever and very interesting.

As a reviewing reader with a background in heavy electrical power engineering, I found the paper scientifically sound and the research generally logical. But perhaps because of my background, I found the paper hard to follow in the following three aspects:

1. The paper starts by talking about renewable energy and environmental effect. But the power level demonstrated is at most in mW (Extended data 7). I can't easily see the relevance of the energy level to the global energy issues. I wish, at least, the authors could show the recovered power level in comparison to the power consumption of the original electronic device. I see the proposed ELRG as a useful energy scavenging mechanism to power some sensor and/or control devices. It could also be exploited to increase the sensitivity of sensing electric field, e.g. detecting whether an electric power wire is live – which is useful in power grid maintenance.

2. I hope the clarity of presentation could be improved. The diagrams are of high quality but they are complicated. For instance, in Figure 2(a), I am not sure whether the arrows are indicating the current direction or the direction of electrons, which should be opposite. I was not able to find Table 1 in 'Extended Data Fig.2'. (Have I missed it in reading?) The permittivity of the liquid dielectric is assumed to be $\epsilon=80$. I think this should be $\epsilon_r=80$. Water is used in experiment. For water, its relative permittivity varies around 80 as the temperature changes below 100 degC. What is the water temperature when the experimental results are compared with COMSOL simulation? $x(\text{mm})$ should be defined in Figure 1 and Extended data 2. Furthermore, although references are provided for TEG, some explanation could help readers who are not familiar with it. For instance, what would cause the triboelectrification material to separate from the first electrode (Figure 2)?

3. I don't quite agree with the rectifier paralleling scheme shown in Figure 4(a). If the original electronic devices are working simultaneously, only the rectifier with the maximum output DC voltage would work - the others are reverse biased.

Reviewer #2 (Remarks to the Author):

As a results of my reaserch the idea seems to be new. The paper is generally well written.

My only question is about the volume/energy estimation compared to other energy harvesting methods and the lifetime and cost of the dielectric if used in such application.

Reviewer #3 (Remarks to the Author):

This paper reports a novel energy scavenging mechanisms namely energy-loss return gate (ELRG), which claims to be capable of harvest electrostatic potential energy and at the same time help to reduce the dielectric loss of the running electronic systems. I found the concept is very interesting and the authors have presented justified review, and a range of work and associated results includes computational simulation, triboelectric generator fabrication, and experimental demonstration (supported by data and videos). This will be of interest to the others in the community. However I felt that the scientific objectives could be tightened up and some part of the presentation could be further clarified to avoid ambiguity in the following areas:

- 1) A real image of the fabricated TEG device could be included in extended data 3.
- 2) What's the real experimental setup for extended data figure 5? TEG? In the schematic, the liquid dielectric covers part of the areas of the electronics device, is that the case for the real set up?
- 3) Extended data 6, only presents charging effect which came from a (iii). Could voltage vs time be measured from a (ii)? How would those results compared with the simulated counterparts as those in extended data figure 1? What might have caused discrepancies?
- 4) The Authors claim e.g. L 85 (an L206 etc.), that inclusion of ELRG in TEG will boost electric power generation. Am not convinced this makes a compelling case to demonstrate the application. More discussion on power densities, efficiencies etc. Could be included. Based on that, perhaps more suitable applications could be depicted.
- 5) The Authors also claim e.g. L 90, that the inclusion of ELRG, can decrease power consumption. This may be true for materials with high thermal conductivities e.g. water, due to cooling effect? However, further results (at least via COMSOL Multiphysics) should be presented to take thermal conductivity of different materials into consideration.
- 6) Water as the liquid dielectric is low cost, easily sourced. However, from demonstrating potential application point of view, the relative dimensions of the water ELRG seem to be very large. I also have concern of placing water adjacent to running electronic/electrical devices. Any discussion on this please? Could results from different types of liquid dielectric be evaluated and hopefully showing promising practical potential, even by simulation?

Please maintain my anonymity to the Authors!

Reviewer #1

- We sincerely appreciate the help of a number of invaluable suggestions/questions concerning the manuscript from reviewer #1, and all of the comments made substantial contributions to our study. Specifically, your comments related to circuit design have guided our understanding of what is impossible from our demonstration. We make no claim that there were some misunderstandings which need to be revised in our study, but we do believe that our revised manuscript represents a substantial improvement over what most researches are currently doing or what most contemporary researchers are currently recommending.

Reviewer #1 Q1 (a) The paper starts by talking about renewable energy and environmental effect. But the power level demonstrated is at most in mW (Extended data 7). I can't easily see the relevance of the energy level to the global energy issues. I wish, at least, the authors could show the recovered power level in comparison to the power consumption of the original electronic device. **(b)** I see the proposed ELRG as a useful energy scavenging mechanism to power some sensor and/or control devices. It could also be exploited to increase the sensitivity of sensing electric field, e.g. detecting whether an electric power wire is live – which is useful in power grid maintenance.

Q1 (a) The paper starts by talking about renewable energy and environmental effect. But the power level demonstrated is at most in mW (Extended data 7). I can't easily see the relevance of the energy level to the global energy issues. I wish, at least, the authors could show the recovered power level in comparison to the power consumption of the original electronic device.

- Before discussing your first comment, we calculated the recovered power level in comparison to the original power consumption and inserted data in the manuscript. This is more informative to readers, and clarifies the descriptions/explanations:

“In this case, recovered power level can be estimated with the formula: $\frac{\text{Recovered power from ELRG}}{\text{Total power consumption}} = \frac{148V \times 125\mu A}{5,110mW} = 0.00359 = 0.36\%$, and power density also can be calculated as: $\frac{\text{Recovered power from ELRG}}{\text{Volume of liquid dielectric}} = \frac{148V \times 125\mu A}{0.001m^3} = 18.583W/m^3$.” **(Revised in the manuscript, page 11)**

- The only experimental data provided in this study may or may not be regarded as a solution to global energy issues. However, the basic motive for ELRG is to recapture every energy loss of an electric system and energy harvesting methods. This means the ELRG process requires no input energy, not disturbing original electric systems. What is especially important about this view is that the ELRG method can easily be integrated without high cost and complex fabrication. If we assume for a moment that ELRG would be integrated in every global electricity demand which is more than 20 000 TWh (referenced from OECD/IEA World Energy Outlook 2009), the total recovered power is approximately 72 TWh. Moreover, as can be seen in Fig. 2d–e, energy harvesting efficiency can be boosted by a power level of 350% and a charging performance of 240%, respectively. Therefore,

our belief is that the ELRG principle can certainly contribute to global energy issues and engineering regimes.

Q1 (b) I see the proposed ELRG as a useful energy scavenging mechanism to power some sensor and/or control devices. It could also be exploited to increase the sensitivity of sensing electric field, e.g. detecting whether an electric power wire is live – which is useful in power grid maintenance.

- Further, your suggestion regarding sensing an electric field is a highly relevant application area. To demonstrate the possibility of a wire sensing method, we conducted lab-scale experiments integrating ELRG in the small electronics and wires. The following discussion of experiments has been included in Fig. 4 with an additional supplementary video attachment:

“In addition, we propose a self-powered wire sensing system with an ELRG mechanism. As illustrated in Fig. 4e, the basic motive for this mechanism is to detect electric signals utilizing electric energy generated by an ELRG. To demonstrate the self-powered mechanism, we scavenge energy loss from electronic devices that are integrated with ELRG. When the electric energy from ELRG is supplied to a sensing system, we can detect electric signals in electrical components. Fig. 4f shows a demonstration of the sensing system with three power lines—ELRG 1 connected to device 1, ELRG 2 connected to device 2, and ELRG 3 connected to device 3. To visualize the sensing system, we integrate light-emitting diodes (LEDs) in the sensing system, which is composed of rectifiers and capacitors (10 μ F) (Extended Data Fig. 10). The respective capacitors are instantaneously charged due to electric energy supplied from the ELRG system, lighting up LEDs connected to the capacitors. When device 1 is turned off, the LED connected to ELRG 1 is turned off while the connected capacitor is discharged. However, when device 1 is turned on, the capacitor is charged, lighting up the LED again (Fig. 4g and Supplementary Video 2).” (Revised in the manuscript, page 12)

Figure 4 ELRG is applicable in our everyday environment. All electric devices are plugged into a 60-Hz AC input electrical source. Detailed electric outputs are given in Extended Data 8–9. **e.** Schematics illustrating self-powered wire sensing conformation. **f.** Optical picture of the self-powered wire sensing ELRG system. Three ELRG outputs are detected by capacitor circuits and LEDs. **g.** Plots showing electric potential variation in the connected capacitors

Reviewer #1 Q2 (a) I hope the clarity of presentation could be improved. The diagrams are of high quality, but they are complicated. For instance, in Figure 2(a), I am not sure whether the arrows are indicating the current direction or the direction of electrons, which should be opposite. **(b)** I was not able to find Table 1 in ‘Extended Data Fig.2’. (Have I missed it in reading?) **(c)** The permittivity of the liquid dielectric is assumed to be $\epsilon=80$. I think this should be $\epsilon_r=80$. Water is used in experiment. For water, its relative permittivity varies around 80 as the temperature changes below 100 degC. What is the water temperature when the experimental results are compared with COMSOL simulation? **(d)** $x(\text{mm})$ should be defined in Figure 1 and Extended data 2. **(e)** Furthermore, although references are provided for TEG, some explanation could help readers who are not familiar with it. For instance, what would cause the triboelectrification material to separate from the first electrode (Figure 2)?

Q2 (a) I hope the clarity of presentation could be improved. The diagrams are of high quality, but they are complicated. For instance, in Figure 2(a), I am not sure whether the arrows are indicating the current direction or the direction of electrons, which should be opposite.

- Our backgrounds in the triboelectric energy harvesting mechanism are reflected in our description in Figure 2a for demonstration of how free electrons flow. The point of this discussion is that when the electrode is charged because of flow of free electrons, the triboelectrification layer is normally negatively charged (such as for a polymer). Thus, once triboelectrification layer contacts with the first electrode, and the first electrode is positively charged due to free electrons that flow toward the second electrode. After that, the electric potential difference between the first and second electrodes drives the stream of free electrons. However, the omitted approach to demonstration in Fig. 2a may be flawed on some points, because there is lack of explanation that describes how an electrode generates electric potential and current flow. To avoid such ambiguities and interpretive difficulties, we have revised our manuscript in Fig. 2a to explain the step-by-step working mechanism with proper schematic description:

“At first, the negatively charged triboelectrification material and the electrode are in an equilibrium state without generating electric charge in the first electrode (Fig. 2a(i)). Due to contact electrification between the triboelectrification material and the electrode, the first electrode becomes positively charged, while the EPD between the first and second electrodes drives the stream of free electrons. As a result, electric current flows between the two electrodes in the direction illustrated in Fig. 2a(ii). After the triboelectrification material separates from the first electrode, current flows in the opposite direction between the two electrodes. It clearly comes as no surprise that both the triboelectrification layer and the first electrode return to an equilibrium state (Fig. 2a(iii)).” (Revised in the manuscript, page 7)

Figure 2 Effects of ELRG are demonstrated through triboelectric generator. **a**. Conformation and process of triboelectric generator. All electric performances are measured under a 9Hz TEG. (i) Equilibrium state of triboelectric generator without liquid dielectric layer; (ii) Contact state without liquid dielectric layer; (iii) separation state without liquid dielectric layer; (iv) Equilibrium state of triboelectric generator with liquid dielectric layer; (v) contact state with liquid dielectric layer; (vi) separation state with liquid dielectric layer.

Q2 (b) I was not able to find Table 1 in ‘Extended Data Fig.2’. (Have I missed it in reading?)

- Table 1 is not included in Extended Data Figure 2. To avoid confusion, we have moved Table 1 to the main manuscript and revised our sentence as Table 1 and Extended Data Figure 2 to be more visible.

Q2 (c) The permittivity of the liquid dielectric is assumed to be $\epsilon=80$. I think this should be $\epsilon_r=80$. Water is used in experiment. For water, its relative permittivity varies around 80 as the temperature changes below 100 degC. What is the water temperature when the experimental results are compared with COMSOL simulation?

- Regarding your suggestion, it should be noted that permittivity of a liquid dielectric, especially water, is highly affected by temperature changes. Thus, we have added the temperature condition of liquid dielectric in Table 1 (20 °C) and revised expression methods which should be relative permittivity ($\epsilon \rightarrow \epsilon_r$). Further, every water temperature condition has been fixed by 20 °C when it comes to both real experiments and COMSOL

analysis. After revising the expression method, we hope to make it clear that we used water at room temperature to demonstrate the practicability of the ELRG method.

Q2 (d) $x(\text{mm})$ should be defined in Figure 1 and Extended data 2.

- As per your suggestion, we added the definition of $x(\text{mm})$, which represents the distance between the first dielectric layer and the ELRG electrode:

“Fig. 1.d shows a plot of the electric potential difference (EPD) between the electric system (V_1) and the ELRG electrode (V_2) as a function of distance (X) which represents the distance between the first dielectric layer and the ELRG electrode.” (Revised in the manuscript, page 6)

Q2 (e) Furthermore, although references are provided for TEG, some explanation could help readers who are not familiar with it. For instance, what would cause the triboelectrification material to separate from the first electrode (Figure 2)?

- Invented by Prof. Z. L. Wang’s group in 2012, a triboelectric generator (TEG) and a triboelectric nanogenerator (TENG) are both energy harvesting methods that convert mechanical energy into electric energy. We might find, for example, that there is always a triboelectric effect when two different materials contact and separate each other. Thus, it is highly possible to recapture mechanical energy in our lives and convert recaptured energy into available energy. To put this discussion in more concrete terms, we provided more discussion in our manuscript about further triboelectric explanation, including the answer to your question, which could help readers who are not familiar with TEG:

“Invented by Prof. Z.L. Wang’s group in 2012, both triboelectric generator (TEG) and triboelectric nanogenerator (TENG) are energy harvesting methods which convert mechanical energy into electric energy²³⁻²⁵. Triboelectrification is a surface charging process in which electrostatic potential is generated by a continuous process of contact and separation between materials. Without neither high cost nor complex fabrication, it can be integrated in our lives to harvest waste mechanical energy. Moreover, the TEG mechanism is also reliable input energy source, because electric output frequency and amplitude can be easily adjusted. For this reason, the TEG is employed as a controllable electric input source to develop strategies for investigating ELRG mechanism.” (Revised in the manuscript, page 7)

Reviewer #1 Q3. I don't quite agree with the rectifier paralleling scheme shown in Figure 4(a). If the original electronic devices are working simultaneously, only the rectifier with the maximum output DC voltage would work - the others are reverse biased.

- After considering your comment about the circuit described in Fig. 4a, we understood what the problem is in our circuit design, and highly regard your opinion. For this reason, we have revised most of data related with the previous Fig. 4a to enhance circuit engineering reliability, and changed paragraphs that were described in the original manuscript to be clear:

“Fig. 4.a illustrates schematically some applications of the ELRG energy harvesting process in our daily lives. As discussed in the above sections, there is unavoidable energy loss from electronic devices such as laptops, cell phones, and computers. To scavenge energy loss from the various electronic devices, we introduce and implement an energy harvesting model, utilizing the ELRG system. Electrical components such as rectifiers, capacitors, and regulator are applied to each device for energy harvesting (Extended data Fig. 7). As a result, with the aid of the ELRG process, it is possible to convert the energy loss to available electricity, which can then be applied to other electric systems. Fig. 4.b shows the demonstration of this application in an actual environment with the following components: ELRG 1 connected to the laptop, ELRG 2 connected to the cell phone, and an energy harvesting circuit. The detailed electric outputs are shown in Extended Data Fig. 8. When an ELRG energy harvesting units is connected to the energy harvesting circuit, the charge storage capacitor (1000 μF) in the circuit is successfully charged by these ELRG power units (Extended Data Fig. 9). As demonstrated in Fig. 4.c, a discharged Bluetooth mouse starts working after connecting it to the charged energy harvesting circuit, operating for a few seconds. Furthermore, as illustrated in Fig. 4.d, a widely used Bluetooth keyboard without a battery can also be operated for tens of seconds by the ELRG system, where a 6000 μF capacitor is used in this experiment (Supplementary video 1).” (Revised in the manuscript, page 11-12)

Figure 4 ELRG is applicable in our everyday environment. All electric devices are plugged into a 60Hz-AC input electrical source. Detailed electric outputs are given in Extended Data 7-9. **a.** Integration of ELRG in individual connection for constructive energy harvesting. **b.** Optical picture of the integrated ELRG system. Two ELRG outputs are connected to other energy harvesting circuits respectively. (from cell phone and lab top) **c.** Discharged Bluetooth mouse is operated by connected energy harvesting circuit. **d.** Discharged Bluetooth keyboard is operated by connected energy harvesting circuit.

Reviewer #2

- We sincerely thank reviewer #2 for the high recognition of our demonstrations. Moreover, we also want to acknowledge the key role provided from your question, regarding material lifetime and cost of dielectric, which must be discussed to guarantee practicability. With your comments, we believe that our demonstration will provide a means for settling the problem of energy harvesting efficiency and durability.

Reviewer #2 Q1. As a results of my research the idea seems to be new. The paper is generally well written. **(a)** My only question is about the volume/energy estimation compared to other energy harvesting methods and **(b)** the lifetime and cost of the dielectric if used in such application.

Q1 (a) My only question is about the volume/energy estimation compared to other energy harvesting methods.

- Perhaps the best way to understand the performance of energy harvesting is a comparison with equivalent input conditions. Because we all know that each energy harvesting method is operated in a different condition, an ideal comparison between energy harvesting methods cannot be achieved. However, it is possible to compare volume/energy estimation of energy harvesting methods with general figures, which is the natural characteristic of the system itself. Therefore, the different volume/energy estimations, including our ELRG method, have been summarized below according to recent energy harvesting research.

	Energy	Volume	Estimation
ELRG	18.583mW~	0.001 m ³	18.583W/m ³ ~
Triboelectric Generator	250 mW ~	0.6 cm ³	416,000 W/m ³ ~
Solar-panel	220W ~	0.058 m ³	3793.103 W/m ³ ~
Wind Generator	1.85MW ~	1099 m ³	1683.348 W/m ³ ~

Triboelectric Generator: Referenced from Zhu, Guang, et al. "Radial-arrayed rotary electrification for high performance triboelectric generator." *Nature communications* 5 (2014): 3426.

Solar-panel: Reference from *HIT Photovoltaic Module with 19.8% cell conversion efficiency*

Wind Generator: Referenced from www.ge.com/wind

- In addition, our ELRG demonstration would not disturb an original electric system while requiring no additional input energy. Thus, because the ELRG mechanism can easily be integrated with every energy harvesting field, the above estimation is not a critical factor in this study. (One example was demonstrated in Fig. 2).

Q1 (b) the lifetime and cost of the dielectric if used in such application.

- Before considering the lifetime and cost of the dielectric, we would like to focus on an infinite natural resource that has received little consideration so far in selecting dielectric materials in the energy harvesting area—water. This may seem to be an onerous task due to restrictions by difficulty of liquid control. However, we demonstrated that it can be integrated into energy harvesting methods without neither complex fabrication or material cost. Unless a liquid dielectric is selected as a high-cost material, we do not need to worry

about problems related to cost. Moreover, because our findings are not highly related with mechanical or chemical activities, the lifetime of the dielectric material would not be threatened.

**“Other advantages of liquid dielectrics are discussed in the introduction of manuscript.”
(Discussed on manuscript, page 3)**

Reviewer #3

- We sincerely want to thank reviewer #3 for the invaluable comments concerning our manuscript, and every insightful comment has helped us sharpen our findings while upgrading the quality of our manuscript. While revising our manuscript, rather than continuing to see our advantages as limited to what we had demonstrated, we carefully observed in terms of what we need to supplement and adjust. Specifically, each of the suggestions/questions serves an important purpose to tighten up scientific objectivity, including essential experimental/analytical factors.

Reviewer #3 Q1. A real image of the fabricated TEG device could be included in extended data3.

- At your suggestion, the real image depicting setup should be noted to enhance experimental reliability. With a TEG image, the development of the figure and manuscript described in revised Extended Data 3 will surely clarify the questions regarding experimental objectivity:

Extended data figure 3 Mechanism of triboelectric electricity generation. **a.** Conformation of triboelectric generator. **b.** Schematic illustration showing step-by-step mechanism of triboelectric generator. (i) Contact state; triboelectric layer and electrode are in charge equilibrium state. (ii) Separation process; first electrode acquires free electrons and second electrode releases free electrons. (iii) Separation state; triboelectric layer and electrode are in charge equilibrium state. (iv) Contact process; first electrode releases free electrons and second electrode acquires free electrons. **c.** Picture of real experimental setup of triboelectric generator with a dielectric loss layer. **d.** Enlarged view of triboelectric generator with ELRG setup.

Reviewer #3 Q2. What's the real experimental setup for extended data figure 5? TEG? In the schematic, the liquid dielectric covers part of the areas of the electronics device, is that the case for the real set up?

- The experimental data described in Extended Data 5 has been designed for investigating the dimension effect of a liquid dielectric with electronic devices, not TEG. To make the experimental setup maximally available, we introduced thin plastic wrap to fix the dimension of the liquid dielectric layer. The revised figure and manuscript described in Extended Data 5 will help readers clarify the questions regarding experimental setup:

Extended data figure 5 Comparison of the ELRG electric outputs under the geometry of the liquid dielectric.
a. Schematic illustration with geometric factors for measuring ELRG outputs. **b.** Image showing real setup under experimental condition of $d=10\text{ cm}$, $h=10\text{ cm}$, and $L=10$. **c.** Plot showing voltage outputs under the consideration of sectional area and volume, with $L=0$. **d.** Plot showing voltage and current outputs under the consideration of distance between the liquid dielectric and electric system.

Reviewer #3 Q3. (a) Extended data 6, only presents charging effect which came from a (iii). Could voltage vs time be measured from a (ii)? **(b)** How would those results compared with the simulated counterparts as those in extended data figure 1? What might have caused discrepancies?

Q3 (a) Extended data 6, only presents charging effect which came from a (iii). Could voltage vs time be measured from a (ii)?

- In the Extended Data 6, we have considered charging performance with voltage vs. time data, which has been included in Fig. 3c:

Q3 (b) How would those results compared with the simulated counterparts as those in extended data figure 1? What might have caused discrepancies?

- For the case of Extended Data 1, we conducted experiments with lab-scale small size liquid dielectric dimensions because some liquids have the characteristic of a volatile solvent with toxicity. Although both experiments in Extended Data 1 and Extended Data 6 were conducted under different experimental conditions, we missed the explanation regarding why there were discrepancies. To clarify this ambiguity problem, we have added experimental setup and explanation in Extended Data 1. Further, the reasons in Extended Data 5 that represents electrical performances under the consideration of sectional area and volume can be confirmed:

Extended data figure 1 ELRG electric outputs under various liquid dielectric properties. Electric device is plugged into a 60Hz-AC input electrical source. **a.** ELRG open-circuit voltage output with four different liquid materials. Dashed black line represents saturation point. **b.** ELRG open-circuit voltage output under different ion concentration. **c.** Image of real experimental setup with sealed liquid dielectric solvent and electric device, and yellow bar representing 3cm-scale. **d.** Image of solvents introduced in the experiments.

Reviewer #3 Q4. The Authors claim e.g. L 85 (an L206 etc.), that inclusion of ELRG in TEG will boost electric power generation. **(a)** Am not convinced this makes a compelling case to demonstrate the application. More discussion on power densities, efficiencies etc. Could be included. **(b)** Based on that, perhaps more suitable applications could be depicted.

Q4 (a) Am not convinced this makes a compelling case to demonstrate the application. More discussion on power densities, efficiencies etc. Could be included.

- Although we believe that the inclusion of ELRG in TEG will boost electric performance, there should be some additional information that must be considered to depict more suitable application design. As per your recommendation, power density and efficiency are the necessary factors that must be considered for depicting an application model. We added some figures and explanations related to power density issues in the manuscript because there are some problems in interpreting TEG practicability. In case of efficiency issues, we have argued that an amplified TEG with an energy-loss return system can amplify electrical peak power by 350%, and charging performance by 240%, with equivalent input energy:

“In this case, recovered power level can be estimated with the formula: $\frac{\text{Recovered power from ELRG}}{\text{Total power consumption}} = \frac{148V \times 125\mu A}{5,110mW} = 0.00359 = 0.36\%$, and power density also can be calculated as: $\frac{\text{Recovered power from ELRG}}{\text{Volume of liquid dielectric}} = \frac{148V \times 125\mu A}{0.001m^3} = 18.583W/m^3$.” **(Revised in the manuscript, page 11)**

- By considering your suggestion regarding our application, we can discover what was difficult for you to compel our application about the inclusion of ELRG in TEG. As per your comments, the example of fabricated TEG shown here may not be a suitable application model because it may increase total mass of the system, which has to be considered for TEG design. However, our integration model, TEG integrated with ELRG, has been the only simple prototype that was heavily designed for the proof-of-principle experiments regarding ELRG.

Q4 (b) Based on that, perhaps more suitable applications could be depicted.

- What is most important about the inclusion of ELRG in TEG is that ELRG might not disturb the original energy harvesting system. Our personal opinion here is that every system of previous well-established TEG is adequate in constructing an ELRG application model, if the ELRG system is integrated as a thin layer while not decreasing power density. Hence, the task in developing a more suitable application is thus available while

operationalizing the conceptual outcome in some way (e.g., to develop a portable thin-flexible liquid dielectric layer).

Reviewer #3 Q5. (a) The Authors also claim e.g. L 90, that the inclusion of ELRG, can decrease power consumption. This may be true for materials with high thermal conductivities e.g. water, due to cooling effect? **(b)** However, further results (at least via COMSOL Multiphysics) should be presented to take thermal conductivity of different materials into consideration.

Q5 (a) The Authors also claim e.g. L 90, that the inclusion of ELRG, can decrease power consumption. This may be true for materials with high thermal conductivities e.g. water, due to cooling effect?

- The suggestion that the inclusion of ELRG can decrease power consumption has been demonstrated with experimental investigation in Fig. 3b. The following plot shows the power consumption of an electronic device under different ambient temperatures. It is possible to operate electric components, especially transistors, more efficiently under lower ambient temperatures. Thus, for decreasing power consumption while operating an ELRG system, a liquid dielectric should be selected as the high thermal conductivity materials that you mentioned. To be clear regarding your question, the following additional explanation about the cooling effect has been discussed in more detail in the manuscript with further references:

Referenced from <https://electronics.stackexchange.com/questions/278935/do-electronic-devices-consume-more-power-when-the-ambient-temperature-is-cold>

“Because some electric components/materials are working more efficiently under lower ambient temperature, these results suggest that the water acts as a liquid dielectric material working as a coolant and lowering the device temperature^{41,42}.” **(Revised in the manuscript, page 10)**

41 Zhang, X., Jouini, W., Leray, P. & Palicot, J. in *Proceedings of the 2010 IEEE/ACM Int'l Conference on Green Computing and Communications & Int'l Conference on Cyber, Physical and Social Computing*. 392-397 (IEEE Computer Society).

42 Maruyama, M., Takahashi, A., Yajima, H., Takeuchi, A. & Yamashita, N. Reducing Electric Power Consumption for Air Conditioning by Improving Temperature Distribution in Telecom Equipment Rooms. (2013). **(Revised in the manuscript, page 15)**

Q5 (b) However, further results (at least via COMSOL Multiphysics) should be presented to take thermal conductivity of different materials into consideration.

➤ From this point of view, we highly agree with your comment that further results with different thermal conductivities should be identified. We have measured additional power consumption data according to several liquid dielectrics with different thermal conductivities (included in Extended Data 6). Hence, these results suggest that the liquid dielectric should be selected as a high thermal conductivity material, such as water, for operating the device more efficiently:

“The reduced power consumption rates regarding various thermal conductivities are discussed in more details on the **Extended Data Fig. 6.c-d.**” (Revised in the manuscript, page 10)

Extended data figure 6 Experiment configuration to demonstrate ELRG effect. a. Experiment pictures. (i) Without liquid dielectric / without charging; (ii) With liquid dielectric / without charging; (iii) With liquid dielectric / with charging. **b.** Plot showing capacitor charging. **c.** Electronic device power consumption depending on various liquid dielectric materials with different thermal conductivity. **d.** Plot showing reduced power consumption when various liquid dielectrics are integrated with ELRG.

Reviewer #3 Q6. (a) Water as the liquid dielectric is low cost, easily sourced. However, from demonstrating potential application point of view, the relative dimensions of the water ELRG seem to be very large. I also have concern of placing water adjacent to running electronic/electrical devices. Any discussion on this please? **(b)** Could results from different types of liquid dielectric be evaluated and hopefully showing promising practical potential, even by simulation?

Q6 (a) Water as the liquid dielectric is low cost, easily sourced. However, from demonstrating potential application point of view, the relative dimensions of the water ELRG seem to be very large. I also have concern of placing water adjacent to running electronic/electrical devices. Any discussion on this please?

- We introduced a relatively large amount of water in our experiments to clearly compare the electric output of the ELRG, and to verify the effect of the liquid dielectric dimension. As you mentioned, there are surely some restrictions that must be solved to enlarge application fields. However, a thin-wide liquid dielectric layer can effectively operate an ELRG system, as shown in Fig. 4, while harvesting sufficient energy in our daily lives. In this regard, it is our strong belief that a small-portable dimension liquid dielectric layer can also be applied that produces sufficient electrical energy.
- Moreover, it is strongly recommended that a fully packaged liquid dielectric layer be designed because placing water near electronic/electrical devices can cause some issues (e.g., water-proof). To avoid this problem, we covered the electronic/electrical device with water-proof materials such as thin urethan film (Fig. 3 and 4), and filled water-proof wrap with water (Fig. 3, Extended Data 1, Extended Data 5, and Extended Data 6). Because a liquid dielectric material does not directly contact with devices, a liquid dielectric material does not affect the operation of the electronic/electrical device.

Q6 (b) Could results from different types of liquid dielectric be evaluated and hopefully showing promising practical potential, even by simulation?

- The principal outputs from different types of liquid dielectrics have been suggested in Extended Data 1 (relative permittivity vs. electric output) and Extended Data 6 (thermal conductivity vs. power consumption). The follow-up results described information that liquid dielectric material should be selected with a characteristic of (a) high relative permittivity to generate high electrostatic potential, and (b) thermal conductivity to reduce power consumption:

Extended data 1 ELRG electric outputs under various liquid dielectric properties. Electric device is plugged into a 60Hz-AC input electrical source. **a.** ELRG open-circuit voltage output with four different liquid materials. Dashed black line represents saturation point.

“To verify the effects of the cooling effect of ELRG, in consideration of various thermal conductivities of liquid dielectrics, we conduct experiments to measure the power consumption while operating the ELRG process. First, Extended Data Fig. 6c–d show the results of reduced power consumption in the electric device with four different liquid dielectric materials: under room temperature (20 °C), water (0.591 W/(m·K)), n-Hexane (0.1203 W/(m·K)), Isopropanol (0.154 W/(m·K)), and Methanol (0.203 W/(m·K)). As the thermal conductivity of the liquid dielectric material increases, the reduced power rate of the ELRG also increases.” (Revised in the extended data and method)

Extended data figure 6 Experiment configuration to demonstrate ELRG effect. c. Electronic device power consumption depending on various liquid dielectric materials with different thermal conductivity. **d.** Plot showing reduced power consumption when various liquid dielectrics are integrated.

- To suggest a promising application model, we have demonstrated that our ELG model can be utilized in a self-powered wire sensing method (illustrated in Fig. 4). However, we acknowledge that the ideal ELRG application model in this study obviously has not been achieved. Several intense efforts (such as water-proof, miniaturization) may be required before ELRG mechanism can be successfully integrated with electronic/electric devices including energy harvesting fields:

“In addition, we propose a self-powered wire sensing system with an ELRG mechanism. As illustrated in Fig. 4e, the basic motive for this mechanism is to detect electric signals utilizing electric energy generated by an ELRG. To demonstrate the self-powered mechanism, we scavenge energy loss from electronic devices that are integrated with ELRG. When the electric energy from ELRG is supplied to a sensing system, we can detect electric signals in electrical components. Fig. 4f shows a demonstration of the sensing system with three power lines—ELRG 1 connected to device 1, ELRG 2 connected to device 2, and ELRG 3 connected to device 3. To visualize the sensing system, we integrate light-emitting diodes (LEDs) in the sensing system, which is composed of rectifiers and capacitors (10 μ F) (Extended Data Fig. 10). The respective capacitors are instantaneously charged due to electric energy supplied from the ELRG system, lighting up LEDs connected to the capacitors. When device 1 is turned off, the LED connected to ELRG 1 is turned off while the connected capacitor is discharged. However, when device 1 is turned on, the capacitor is charged, lighting up the LED again (Fig. 4g and Supplementary Video 2).” (Revised in the manuscript, page 12)

Figure 4 ELRG is applicable in our everyday environment. All electric devices are plugged into a 60-Hz AC input electrical source. Detailed electric outputs are given in Extended Data 7–9. **a.** Integration of ELRG in individual connection for constructive energy harvesting. **b.** Optical picture of the integrated ELRG system. Two ELRG outputs are connected to an energy harvesting circuit. (from cell phone and laptop) **c.** Discharged Bluetooth mouse is operated by connected energy harvesting circuit. **d.** Discharged Bluetooth keyboard is operated by connected energy harvesting circuit. **e.** Schematics illustrating self-powered wire sensing conformation. **f.** Optical picture of the self-powered wire sensing ELRG system. Three ELRG outputs are detected by capacitor circuits and LEDs. **g.** Plots showing electric potential variation in the connected capacitors

Reviewer #1 (Remarks to the Author):

Thanks you for providing the report detailing the revision carried out.

The revised paper is much easier to read. I believe the readers can now quickly understand the purpose of the development, method of the study and working principles involved in the demonstration. I think the paper can be accepted as it is. But it would be useful if the paper could further include a very brief discussion or indication of the dependency (or independency) of the liquid permittivity on the frequency of the electric field. This would allow the readers to judge the suitability of the selected liquid dielectric in different applications.

I would suggest some very minor modifications in the early part of the paper to help the reader a bit further. But these are not suggested as mandatory changes.

1. Line 57. 'electronic devices.' could be modified as 'electronic devices, whose power supply may need to support autonomous operation over long periods of time to reduce the carbon footprint.'
2. Line 63. 'Electric fields,' could be modified as 'Alternating electric fields,'
3. Line 76. 'To minimise the energy loss arising from dielectric loss' could be modified as 'To minimise the dielectric loss'
4. Line 128. 'simulations'. Could this be '2D simulations'?
5. Table 1. 'liquid dielectric'. Could this be simply 'de-ionised water'?

Reviewer #3 (Remarks to the Author):

the authors have addressed key issues raised.

Reviewer #1 The revised paper is much easier to read. I believe the readers can now quickly understand the purpose of the development, method of the study and working principles involved in the demonstration. I think the paper can be accepted as it is.

- We would like to express our sincere gratitude for your kind consideration and high recognition of our manuscript.

Q1) But it would be useful if the paper could further include a very brief discussion or indication of the dependency (or independency) of the liquid permittivity on the frequency of the electric field. This would allow the readers to judge the suitability of the selected liquid dielectric in different applications.

- As your suggestion, we highly acknowledge that the dependency of the liquid permittivity on the frequency of the electric field is necessary factor, which helps researcher select a suitable liquid dielectric for ELRG operation. In general case, the dielectric constant is decreased with an increase of the input frequency. If input frequency is faster than the relaxation time which is defined as a time to be properly aligned with the field, the dipoles would not be properly aligned. Therefore, we added the explanation for your suggestion.

“In general cases, the dielectric constant is decreased as the input frequency is increased. In this study, the ELRG is investigated in the very low frequency range, up to 60Hz. Thus, we assume that the liquid dielectric constant is not reduced by input frequency for the ELRG process.”
(Revised in the method section of manuscript, page 14)

Q2) I would suggest some very minor modifications in the early part of the paper to help the reader a bit further. But these are not suggested as mandatory changes.

- We carefully checked your comments regarding modifications in our manuscript and revised the manuscript as your opinion except 1st suggestion. In our opinion, in case of 1st suggestion, it is better to maintain our original sentence considering entire context. We sincerely thank to your kind attention in our manuscript.

1. Line 57. 'electronic devices.' could be modified as 'electronic devices, whose power supply may need to support autonomous operation over long periods of time to reduce the carbon footprint.'

2. Line 63. 'Electric fields,' could be modified as 'Alternating electric fields,'

3. Line 76. 'To minimise the energy loss arising from dielectric loss' could be modified as 'To minimise the dielectric loss'

4. Line 128. 'simulations'. Could this be '2D simulations'?

5. Table 1. 'liquid dielectric'. Could this be simply 'de-ionised water'?

Reviewer #3 the authors have addressed key issues raised.

- We would like to express our sincere gratitude for your kind consideration and high recognition of our manuscript.